# Risk and Resilient Functioning of Families of Children with Cancer during the COVID-19 Pandemic

**DOI:** 10.3390/ijerph20065208

**Published:** 2023-03-16

**Authors:** Renee Gilbert, Carolyn R. Bates, Devanshi Khetawat, Meredith L. Dreyer Gillette, Rachel Moore

**Affiliations:** 1Clinical Child Psychology Program, Dole Human Development Center, University of Kansas, Lawrence, KS 66044, USA; 2Department of Pediatrics, University of Kansas Medical Center, 3901 Rainbow Blvd, Kansas City, KS 64114, USA; 3University of Kansas Cancer Center, 4001 Rainbow Blvd, Kansas City, KS 64114, USA; 4Department of Pediatrics, Children’s Mercy Kansas City, 2401 Gillham Rd, Kansas City, MO 64108, USA

**Keywords:** cancer, pediatrics, psycho-oncology, family, COVID-19, risk, resilience

## Abstract

Previous literature highlights the impact of COVID-19 on family functioning. Less is known about the impact of the pandemic on families of pediatric cancer patients. In order to determine universal and unique risk and resilience factors of these families during the pandemic, a qualitative analysis was conducted on families currently receiving cancer treatment at a Midwestern hospital. Results of the data analysis depict ways in which these families have been impacted by and have adapted to COVID-19. These findings suggest that families of pediatric cancer patients have unique experiences in the context of COVID-19, in addition to universal experiences outlined in previous literature.

## 1. Introduction

The COVID-19 pandemic presents several unprecedented challenges for children and adolescents [1]. Specifically, it has had significant ramifications for families of youth with chronic pediatric health conditions. Global health and economic uncertainty, alongside state and local stay-at-home orders during the COVID-19 pandemic, resulted in substantial changes to daily life [2]. For instance, the pandemic led to several alterations in the family home environment and organization [3]. These significant changes could increase susceptibility to psychosocial risk, as well as open opportunities for resilience among families with children and adolescents [4]. Of note, families of children undergoing cancer treatment may have experienced unique challenges and stressors during the COVID-19 pandemic, as children undergoing chemotherapy were at high risk for COVID-19 infection [5,6,7]. Given the importance of family functioning on child and caregiver psychological outcomes during pediatric cancer treatment [8], and the direct role of the family during the COVID-19 pandemic [4], it is important to understand the unique experiences of families of children with cancer during the COVID-19 pandemic to promote long-term psychosocial wellbeing among children with cancer and their families throughout public health crises.

Previous literature provides evidence of mixed outcomes when exploring the psychosocial outcomes of COVID-19 among families of pediatric cancer patients. On the one hand, some studies suggest that the social disruptions that emerged in the wake of COVID-19 may have a negative impact on family functioning and routines [4]. Changes in daily activities and familial roles that accompany childhood cancer can also function as added stressors [9]. Thus, given the increased susceptibility to COVID-19 among youth with cancer, heightened disruptions to resources and support, and increased financial demand, negative psychosocial outcomes can be anticipated [10,11,12]. An Italian study found that caregivers reported clinically significant levels of post-traumatic stress, acute stress, and anxiety during the pandemic [13]. A British study found that caregivers no longer viewed the hospital as a “safe place” and generally endorsed feeling highly worried about the pandemic and vigilant [14]. Caregivers in an American study reported disruptions in anxiety, sleep, eating habits, mood, exercise, stress, and financial status [15]. Hence, globally, the COVID-19 pandemic has been associated with negative psychosocial outcomes for families with children and adolescents with cancer.

Despite these findings, resilience has also been demonstrated in families of children and adolescents with cancer during the COVID-19 pandemic. Prior to the COVID-19 pandemic, families generally displayed resilience in the face of a cancer diagnosis and treatment [16,17]. Emerging literature, however, suggests that these families may be uniquely resilient in the face of the COVID-19 pandemic. A Dutch study found reduced levels of caregiver distress following the advent of the pandemic [18]. Another study in the U.S. found that caregivers generally reported adjusting well to the pandemic [19]. One reason for these positive findings may be that families of pediatric cancer patients are “uniquely prepared” for the COVID-19 pandemic. Indeed, families may have taken precautions that mimic COVID-related measures prior to the pandemic (e.g., wearing masks and social distancing) [19]. Additionally, decreased social and job demands, increased family time, and access to education technology and healthcare may lead to a reduction of stress in these families [20]. Processes like meaning making, flexibility, connectedness, social and economic resources, and collaborative problem solving may foster familial resilience [21]. Previous work has found that greater family cohesion, expressiveness, and support were associated with better psychosocial outcomes for both pediatric patients and their siblings [17]. While certain findings currently demonstrate evidence of increased challenges for these families, there is also evidence of resilience and adaptive adjustment in the face of COVID-19. Thus, an in-depth exploratory study is needed to investigate the presence of risk and resilience in these families. 

Prime and colleagues [4] developed a conceptual framework derived from literature on family systems theory [22,23], Bronfenbrenner and Morris’ bioecological model [24], a family stress model [25], and developmental systems theory [24]. The authors suggest a cascading influence of the COVID-19 pandemic on child adjustment in which social disruptions lead to heightened distress among caregivers, impacting dyadic relationships within the family (marital, parent-child, siblings). This may in turn impact family processes that influence child adjustment. This model is a “mutually reinforcing system” in which stress in one domain may lead to stress in another [4]. Prime and colleagues have identified familial processes and individual factors that may increase risk or promote resilience during the COVID-19 pandemic. Indeed, pre-existing vulnerabilities (e.g., economic hardship, racism, and marginalization) may serve as risk factors for families, increasing vulnerability to social disruptions and other COVID-19 related complications, whereas family well-being (e.g., good communication, organization, and strong belief systems) may increase resilience or serve as a protective factor against these disruptions [4]. Less is known about whether or not this theoretical framework aligns with families of pediatric cancer patients. Findings of the present study will identify unique risk and protective factors to best inform clinicians who work with these families. 

### Purpose of the Current Study 

Guided by the theoretical framework proposed by Prime et al., this qualitative study aims to describe the psychosocial risk and resilience among families of children who were receiving active cancer treatment during the onset of the COVID-19 pandemic. In order to support and promote well-being in families of pediatric cancer patients throughout the COVID-19 pandemic and during future public safety hazards, it is important to identify areas of resilience and potential areas of vulnerability. 

## 2. Materials and Methods

This study is part of a larger study that began prior to the COVID-19 pandemic that sought to examine changes in the family home environment following a new diagnosis of pediatric cancer [26]. A subset of study data was collected during the COVID-19 pandemic, and these data were examined separately from the larger dataset given the impact of the COVID-19 pandemic on the family home environment. Families were recruited from a large children’s hospital in the Midwestern U.S. Primary caregivers of children who met the following eligibility criteria were identified via the institution’s clinical research database: (1) between the age of 18 months and 17 years, 11 months, (2) diagnosed with cancer within the past year, (3) living within the home of a legal guardian and not in foster care, and (4) undergoing active treatment (i.e., chemotherapy, radiation, or bone marrow transplantation). Participants were deemed ineligible if their child was diagnosed with a recurrence or secondary malignancy. Primary caregivers also needed to be English-speaking to complete semi-structured interviews. With primary medical team approval, participants were approached by study personnel during outpatient oncology clinic appointments or remotely via phone to conduct informed consent and administer study measures. The hospital’s Institutional Review Board approved all study procedures. The authors report there are no competing interests to declare. 

A total of 31 parents (24 mothers and 7 fathers) participated in the qualitative portion of the main study. The mean age of children was 6.6 years old (SD = 4.21) and ranged between 18 months to 16 years. Females made up 50% of the sample of children. The sample was primarily white (78%). Families had an average of 4.90 people living in the home (SD = 1.46) with a range of 3-9 individuals. Fifty-six percent of the sample had a diagnosis of leukemia, and average time since diagnosis was 9.18 months with a range of 1.50–20.83 months. Table 1 displays the demographic characteristics of the sample. 

The parent study began data collection prior to the onset of COVID-19 to evaluate the influence of a cancer diagnosis on the home environment. However, the data presented in the current analysis were collected after the onset of the COVID-19 pandemic between July 2020 and January 2021. Following the onset of the pandemic, all families in the parent study were contacted to complete additional COVID-19 measures, regardless of whether or not they completed the initial data collection prior to the pandemic. The current sample is a subset of families who agreed to be interviewed remotely via phone. Additional COVID-19 measures have been published elsewhere [26].

### 2.1. Measures: Qualitative Assessment

Prior to the COVID-19 pandemic, a 12 item semi-structured interview was created to examine differences in family functioning and the home environment following a new diagnosis of cancer [26]. These initial questions were developed from a review of literature consisting of models of family functioning in response to stress, including the Family Stress Model and Ecological Systems Theory. Families responded to open-ended questions regarding family rules and routines before the diagnosis, and changes or lack thereof following the diagnosis. Following the onset of the COVID-19 pandemic, additional questions were added where caregivers were asked to specify changes that occurred for the family since the pandemic. These questions were added to capture the supplemental impacts of the pandemic on family routines following a new pediatric cancer diagnosis. The interview questions can be seen in Appendix A.

### 2.2. Analytic Plan: Directed Content Qualitative Analysis

All semi-structured audio recordings were de-identified, transcribed, and double-checked for accuracy by trained data transcriptionists. Interviews then were imported into Dedoose software, version 8.0.25 for data management and analysis. Data coding and analysis were conducted by authors RG and DK with oversight by CRB and additional input from senior author RM. The authors utilized directed content analysis. This approach is used to validate or extend a previously established framework [27]. Coding categories are derived from that framework and used to guide qualitative analysis. Guided by Prime and colleagues’ model of risk and resilience of family well-being during the COVID-19 pandemic, the pre-determined categories were as follow: social disruptions, organization, pre-existing vulnerabilities, belief systems, communication, and psychological and physical well-being [4]. Coders (RG, DK) utilized this framework to create rules for these categories and independently code each interview separately. RG and DK compared codes and resolved discrepancies through negotiated agreement with additional input from CRB and RM (see [28,29]). Following this, coders RG and DK, in addition to CRB and RM, reviewed each pre-determined category for various themes that emerged. Themes were reviewed by coders RG, CRB, and RM, and discrepancies were resolved through negotiated agreement.

## 3. Results

Throughout the interviews, parents discussed the impacts of pediatric cancer, COVID-19, and ways of adapting that fell into the pre-determined categories outlined by Prime and colleagues [4]: organization, social disruptions, pre-existing vulnerabilities, communication, belief systems, and psychological and physical well-being. Within each category, various subthemes emerged that depicted the experiences of these families. 

### 3.1. Organization

Family organizational processes, including rules, routines, and rituals, were disrupted by both the child’s cancer diagnosis and the COVID-19 pandemic. Families discussed that treatment-related tasks (e.g., clinic appointments and medication administration) and infection control measures (e.g., social distancing and handwashing) became the top priority for the family both before and during the COVID-19 pandemic. Thus, other rules, routines or rituals that conflicted were often adjusted or relaxed. Many families presented with resilience and adapted by implementing new rules or routines around cleanliness and infection control to protect their immunocompromised child even prior to COVID-19 pandemic, and discussed re-engaging these strategies during the onset of the pandemic: 


*“I said something like having the kids at first, we were very strict about them coming home and changing clothes and even family members, making sure they were washing hands and stuff as soon as they walk in doors and stuff like that. Once she progressed on through her treatment, we got a little more relaxed on some of that. Course now, COVID happened some of that stuff has picked back up and we’re a lot stricter on things again.”*
(Father of 3-year-old)

Families acknowledged the challenges around social distancing and infection control that limited their contact with extended family and friends, which served as a risk factor for insufficient social support during the COVID-19 pandemic. Yet families also displayed resilience in ways of creatively staying in touch with social supports (e.g., connecting through social media; socially distanced meal delivery). Given infection control restrictions during the COVID-19 pandemic, families discussed appreciation for continued access to their hospital’s social worker to provide practical resources, such as gas cards to support transportation to and from the hospital.

Families discussed several additional resilient adaptations in the face of pandemic-related disruptions, including savoring increased togetherness through rituals such as eating meals together when everyone in the family was at home, which promoted family bonding. Parents expressed gratitude for additional caregivers working from home and able to provide back-up childcare for siblings when needed (e.g., when one parent was needed at the hospital). Moreover, increased access to classroom and school material through the provision of universal online schooling supported academic resilience for the ill child, though families also acknowledged the challenges of completing schoolwork on days when their child was not feeling well in the context of many other demands.

### 3.2. Social Disruptions

Families reported many disruptions to their social context during the COVID-19 pandemic. Families acknowledged interactions between pandemic-related isolation alongside aspects of the cancer diagnosis that interfered with social functioning. These included things such as extensive hospital stays, which had been disruptive prior to the pandemic, and during the pandemic also included restrictions on the number of visitors and caregivers. Similarly, increased susceptibility to illnesses and impairing illness-related symptoms (e.g., fatigue) were present prior to the pandemic and limited engagement in some social activities. However heightened risk of infection in social settings during the pandemic led to more severe restrictions on engagement in extracurriculars and physical activities, including sports, as well as routine family outings, attendance at church, school, and even one-on-one social interactions with peers or friends. Families demonstrated resilient adaptation through shifting to online formats for school, church and even family gatherings (via televideo).


*“Well, personally, we attended church until now. But I think it was more social distance than her diagnosis because we were still going to church and everything.”*
(Mother of 16-year-old) 

Families reported various risks and consequences of pandemic-related social disruptions. For example, school closures alongside demands of cancer treatment were reported to result in increased screen time and decreased structured schedules for many children. Families also reported concerns for increased behavioral difficulties and socially inappropriate behaviors due to isolation. Parents further reported financial strains due to loss of work during the COVID-19 pandemic. Families discussed limited spousal support, in addition to decreased support from extended family members that they relied on during cancer treatment prior to the onset of the pandemic. They reported that extensive hospital stays had an increasingly negative impact on their child due to visitor restrictions and isolation from family members: 


*“[She’s] having to stay in the hospital for 5 days when she was diagnosed and just the amount of mental health stress that put on her, not being near her brothers and sisters.”*
(Mother of 7-year-old)

Importantly, some families also reported more minimal impacts of social disruptions on their emotional and behavioral health. For example, one family reported little-to-no disruptions in their family routine. Other families reported that COVID-19-related-isolation had a positive impact due to increased family time at home. Finally, few families reported that a shift to work-from-home gave parents increased availability for cancer treatment obligations (e.g., attending clinic appointments). Some families noted that the children were happy to not go to school, and even reported appreciation for the COVID-19 pandemic: 


*“I hate to say it, but the coronavirus probably has helped us get through a lot of that a lot of this as a family because unfortunately it caused us to all be home... there’s just a lot of things we didn’t have to worry about ‘cause we’re just stuck at home anyway you know. My husband hasn’t been to work, he just works from home now... I think it would’ve looked a lot different if it weren’t for coronavirus because he would’ve been going to work, my other child would’ve been going to school participating in activities. So, the coronavirus has probably helped support our future because it’s given us all you know a reason that we were all at home together through it all.”*
 (Mother of 9-year-old) 

### 3.3. Pre-Existing Vulnerabilities

Several psychosocial, familial, and environmental vulnerabilities which pre-dated the cancer diagnosis served as risk factors toward psychosocial functioning during treatment in the COVID-19 context. Living away from the treatment location was a risk factor as families indicated that they found it difficult to develop the support network for immediate emotional, relational, and physical support in their area when they lived far away from the hospital. For instance, families reported not knowing about local resources or parent groups with a child having a cancer diagnosis: 


*“So that’s been our biggest challenge is finding all the support that we need when we’re not in Kansas City cause when we’re Kansas City it’s much easier. Like there’s Ronald McDonald, like there’s parents’ parents, there’s places who’ll buy toys for the kids if they are bored, there’s all this stuff but now that we’re not in Kansas City and we’re home there’s not any of that. so, finding that outright support has been difficult.”*
 (Mother of 3-year-old) 

Many families also suggested that having complex family dynamics and multiple children at home was a risk factor for worse psychosocial functioning as parents needed to balance out the needs of all their children, and not just the one who had cancer. Complex or co-parenting plans added challenges as the rules and organizational structures varied across homes, making it difficult to maintain consistency of care and expectations. Families further indicated that limited financial income or support could function as a risk factor. Whereas some families expressed gratitude for having commercial insurance or additional sources of income, others expressed benefit of having one stay-at-home parent to manage treatment and daily activities. Additionally, environmental factors, such as cleanliness and living standards caused additional concerns: 


*“Well right after... [he] was diagnosed, we had the home tested for mold ‘cause you can’t live in an environment where there is mold... Our home tested positive for mold. So, they started tearing things out to see where things were at, and it was everywhere. So right after he was diagnosed, we couldn’t go home.”*
(Mother of 3-year-old)

### 3.4. Communication

For families of pediatric cancer patients, communication within families and to extended networks of family and friends was important during this time. Within families, participants report having multiple conversations with the child’s siblings, extended family members, and visitors about the importance of infection control practices to promote safety for the ill child. This included examples such as not going outside without masks, not touching surfaces unnecessarily, sanitizing everything including hands, not hugging strangers, and taking showers and washing hands constantly. Interestingly, families reported that the onset of the COVID-19 pandemic often made it easier for them to explain the germ policies to strangers, siblings, and other family members.

For families of pediatric cancer patients, communication was not only important within families but with their medical teams as well. Families in the present study discussed the importance of clear guidance from providers, but few shared the influence of the COVID-19 pandemic on this communication.

### 3.5. Belief Systems

Resilient family belief systems were evident through discussions of child-rearing, shared responsibilities, family cohesion, and faith. Cancer treatment in the context of the pandemic challenged some parents’ previous beliefs and values about parenting. In order to adapt, parents discussed shifting their views to a broader perspective: 


*“…probably we let a lot more of the little things fly. Don’t get on her as much for small things. You know if she wets her pants or knocks stuff over you know we just try to let little things fly that aren’t going to matter you know in 5 hours from now.”*
(Mother of 4-year-old)

Parents, however, also relied on previous beliefs and expectations for their children during the COVID-19 pandemic and cancer treatment. Families discussed the importance of keeping age-appropriate expectations for behavior during the pandemic and cancer treatment so that the child develops resilient patterns for behavior and resilience after this temporary period of time. Families also discussed beliefs about the importance of keeping normalcy in the household throughout the COVID-19 pandemic and cancer treatment: 


*“... we needed to keep some sense of normalcy. You know there’s—I knew there’s things in my house that I can control and cancer I can’t control.”*
 (Mother of 3-year-old) 


*“…we’re trying to stay as close to normal, to normalcy as possible. But we’re trying to, for everybody at home to have as close to a normal life as possible.”*
(Mother of 2-year-old)

Finally, families shared a reliance value of faith and family strength to endure the COVID-19 pandemic, cancer, and related challenges: 


*“Teaching her how to be a warrior like I was doing that before we got cancer so when we know we got cancer now we’re [about] to beat this too. It’s just a culture in our house that we’re strong, we’re faithful, we accomplish things, we conquer, and I think that would be something that I would teach someone else that maybe doesn’t have that or hadn’t been doing it for their kid.”*
(Mother of 7-year-old)

### 3.6. Psychological and Physical Well-Being

Families reported psychological and physical impacts due to the COVID-19 pandemic and their child’s cancer diagnosis and treatment. Families discussed numerous treatment-related risks to the child’s physical well-being, including fatigue, low energy, physical limitations, and enduring intense procedures. Families also noted their child’s risk through increased vulnerability to illnesses, such as the flu and COVID-19, due to being immunocompromised. Families discussed increased accommodations for the child due to cancer and treatment.

Parents reported negative impacts of the COVID-19 pandemic on their mental health as well. They discussed increased anxiety, stress, and difficulty accessing the social support they needed: 


*“Right now, what has made it harder is like for example my husband and I do everything together. And now we’re having to split because of COVID and that has been very hard for me to go through this without him here.”*
(Mother of 2-year-old)

Families also discussed feeling a heightened sense of distress and overwhelm: 


*“...you just feel like you’re just constantly trying to get your head above water, catch your breath and figure something out and then you get pushed back under. So, it’s like we’re trying to understand what he has going on and what his cancer means and what our life is going to mean.”*
(Mother of 3-year-old)

Some families reported appropriate access to resources to address mental health needs. These resources include individual therapy, couples therapy, and Facebook groups, and they promoted resilient coping. However, many families reported a need for additional resources: 


*“...there’s no, there’s no one to talk to about it. There’s like—besides going in and seeing a therapist—there’s not really a lot where you can be like this is really challenging as a parent...”*
(Mother of 3-year-old)

Importantly, families reported that over time, they experienced reductions in their sense of distress: 


*“I’d say now that um we’re not as stressed because now that he entered his maintenance phase of treatment, and he seems to be doing—like—he acts normal and seems to be doing normal... I do feel like honestly other than when we go to the hospital and stuff everything seems just kind of normal to me... I wouldn’t say there’s a lot of high stress related to his diagnosis anymore.”*
(Mother of 9-year-old) 

## 4. Discussion

The present study sought to investigate risk and resilience factors among families of pediatric cancer patients during the COVID-19 pandemic. The study was guided by the theoretical framework developed by Prime and colleagues [4] to understand universal and unique impacts of the pandemic on these families. Findings suggest that COVID-19 has impacted families in several areas, and additional risk and resilience factors during this time exist in pediatric cancer populations. 

In accordance with Prime and colleagues’ theoretical framework, families in the present study reported distress due to pandemic-related social disruptions. Families reported that the COVID-19 pandemic led to several social disruptions, including school closures, job loss, decreases in extracurricular activities, decreases in support and resources, and isolation from extended family and friends. Families reported that these disruptions often served as risk factors for heightened parental distress due to lack of social support. These findings align with the Prime framework [4], in which they suggest that social disruptions may negatively impact caregiver well-being, family processes, and child adjustment. Indeed, previous work suggests that increased COVID-19 disruptions to daily life is associated with higher levels of familial psychosocial distress [3].

These disruptions, though, have important implications and consequences for families of pediatric cancer patients. Caregiver distress is common during the early phases of child cancer diagnosis and treatment [30], and mental health resources are critical to mitigate the impact of this distress on long-term functioning. For families in the present study, pandemic-related disruptions to social support and mental health resources may increase risk for prolonged psychological distress. Indeed, a perceived lack of social support led to increased psychological impairment among caregivers of youth with cancer in a pre-COVID-19 sample [31,32], and thus future work should consider long-term outcomes of pandemic-related restrictions to resources on parent psychological functioning. In addition to parent support, psychological interventions for caregivers and siblings of children with cancer may teach important skills for coping during the early phases of pediatric cancer treatment [33,34]. Thus, a lack of social support and mental health resources due to the pandemic, as reported by these families, may have negative implications for caregivers, children, and siblings. Notably, some families reported resilience in the face of COVID-19-related disruptions, namely that they have more time to spend with family. These findings also align with previous work noting that since the onset of COVID-19, parents report more time spent in activities with children, in addition to feeling elevated levels of closeness and warmth, compared to the period prior to the pandemic [35]. Indeed, continued engagement in family rituals may have protective functions for children with cancer [36]. However, the long-term impact of this period of isolation on families experiencing acute stressors of pediatric cancer treatment merits ongoing investigation and potentially intervention to mitigate risks and promote lasting resilience. 

Pediatric cancer patients may experience limitations to their autonomy and mobility during intense therapy [37]. COVID-19 restrictions resulted in an additional barrier for children and families to sports and other physical activities. Yet physical activity may have mental health benefits for children with and without cancer, including increased self-esteem [38,39]. Additionally, specific benefits of physical activity for pediatric cancer patients are notable, including increased quality of life, sleep efficiency, and decreased fatigue [40,41]. Thus, pandemic-related disruptions to physical activity may serve as risk factors for children with cancer.

Prime and colleagues also theorized that pre-existing vulnerabilities (e.g., economic difficulties, health conditions, trauma history, racism, and marginalization) would amplify risk-related consequences of the pandemic for families [4]. In our sample, a pediatric cancer diagnosis itself may represent a pre-existing vulnerability. Indeed, for some families, the cancer diagnosis led to social isolation, school dismissal and limited physical activity prior to COVID-19. Additionally, parents reported cancer-related disruptions including lengthy hospital stays and demanding treatment.

Parents reported additional pre-existing vulnerabilities, including socioeconomic status, location, and race. Sociodemographic factors may place a subset of families with a pediatric cancer patient at risk for increased levels of distress [42]. Families reported various difficulties due to their geographic location, including a lack of resources and long distance from the treatment center. Previous literature suggests that geographic areas may have implications for cancer-related health outcomes; in fact, Carriere and colleagues [43] suggest that rurality leads to lower survival rates of cancer, potentially due to lower engagement in treatment and screening. Additionally, moving to a new location for treatment may lead to social stress. For example, families of color reported worries about racism as they moved to a new location for treatment. The COVID-19 pandemic amplified the healthcare disparities for people of color, coupled with heightened racial distress due to the murder of George Floyd [44]. Racism may have negative mental health consequences and physical health consequences [45]. However, further investigation is warranted to evaluate how race-related stress, particularly during the height of the COVID-19-pandemic, may impact outcomes of pediatric cancer patients and families. 

Pediatric cancer also places a financial burden on families (risk-factor), and the COVID-19 pandemic may increase financial difficulties for these families. Expensive treatment, caregiver medical leave, travel, and lodging costs and other nonmedical expenses contribute to the financial burden of cancer [46]. Further, for some families already at a socioeconomic disadvantage (e.g., low-income, single-parent household), a cancer diagnosis and treatment may exacerbate these concerns [42]. The COVID-19 pandemic exacerbated financial difficulties for families of pediatric cancer patients through job loss and increased lodging and transportation costs [47]. Further, pediatric cancer-induced financial burden may worsen parental distress already experienced due to the cancer diagnosis [48].

Families in the present study adapted to the COVID-19 pandemic and cancer through systems of organization and communication. Previous work emphasizes the importance of consistent family rules and routines during times of stress, including after a cancer diagnosis [26] and during the COVID-19 pandemic [3]. However, families in the present study reported many COVID-related disruptions to rules and routines. Indeed, families reported ridding of set schedules for their children due to school closures, relaxing previous rules set (e.g., bedtime), increased screen time and decreased family time (e.g., family dinner). Parental accommodation and laxness are associated with emotional, behavioral and sleep difficulties in children with cancer [49,50]. Families, however, displayed resilience through the development of new rules and routines to adapt to the COVID-19 pandemic and the cancer diagnosis. In their model, Prime and colleagues emphasize organization (i.e., access to resources, connectedness, and adaptability) as an important family process that might be disrupted, or act as a source of resilience, during the COVID-19 pandemic [4]. Many families in the present study reportedly engaged in and prioritized routines regarding cleanliness and infection control, including social distancing, isolation, and sanitization practices and some families reported feeling uniquely prepared for the pandemic, given they already engage in these practices. Families also discussed additional ways in which they adapted to pandemic-related disruptions, through alternating or changing schedules and reliance on additional family members. 

Prime and colleagues also note the importance of communication within families during the COVID-19 pandemic, described as “clear information, emotional sharing, collaborative problem-solving, dyadic and family coping” [4]. In the present study, families emphasized the importance of communicating with siblings and other family members regarding safety protocols for cleanliness and infection control. Handwashing and isolation protocols for children with cancer are generally in line with preventative measures for the general population during the COVID-19 pandemic, which led to unique preparedness for families of pediatric cancer patients [51]. Families also reported the importance of medical team communication throughout cancer treatment, although they did not discuss the influence of the COVID-19 pandemic.

Prime and colleagues [4] note that the impact of COVID-19 on families is seen through a “mutually reinforcing system”, in that stress and disruption in one domain may produce stress and disruption in another. Indeed, families in the present study reported several interactions between domains. For example, increases in social disruptions (e.g., school closures, extracurricular cancellations) led to disruptions in organization (e.g., relaxing of rules and routines). Social disruptions, like job loss, led to financial strain for some families; financial strain serves as a pre-existing vulnerability, which exacerbates the negative impact of the COVID-19 pandemic. These pre-existing vulnerabilities, such as financial strain, and social disruptions, such as job-loss, led to decreases in caregiver well-being. Social disruptions also limited access to mental health resources, which led to decreases in caregiver and sibling well-being. On the other hand, positive adaptation in one domain also led to positive adaptation in another for families in the present study. For example, strong family beliefs about child-rearing practices led to better organization around rules and routines during this time.

### 4.1. Limitations

This study has several limitations to acknowledge. First, the hospital at which this study was conducted serves a population that is predominantly White and non-Hispanic, along with the sample in the present study. This limits our ability to capture cultural differences in family functioning in the context of COVID-19 and cancer, which may have important implications for treatment. Barriers exist that limit recruitment and engagement of marginalized communities in research studies, and it is important that these barriers are addressed in future work [52]. Additionally, caregivers in our study were highly educated and married, and thus our findings may not adequately capture families with less education and single-parent households. Given that our data capture the impact of the pandemic only at one specific time point (i.e., first year of the pandemic), more work is needed to understand the impact of the pandemic on families of children receiving cancer treatment. Furthermore, the caregivers in our study were predominantly female. Although previous literature suggests mothers and fathers do not differ in affective responses, anxiety, post-traumatic stress symptoms or quality of life in the face of childhood cancer [30,53,54], some literature suggests that mothers cope more effectively and report higher rates of coping than fathers given a diagnosis of childhood cancer [55]. Fathers are historically under-represented in pediatric research [56], and future work should address barriers to ensure father’s perspectives are taken into account. Finally, given our limited sample and single-site study, we did not report frequencies of participants who reported various themes as we felt these statistics may not necessarily generalize to other families of youth with cancer. Future studies with larger sample sizes may consider reporting these statistics.

### 4.2. Clinical Implications

Results of the present study have crucial clinical implications for families of pediatric cancer patients during the COVID-19 pandemic. The Prime model [4] emphasized the importance of mental health interventions for families during this time, citing universal psychosocial consequences that may result from the disruptions of the pandemic. Our present study, however, has implications for the unique needs of these families who endured the first year of the COVID-19 pandemic during their cancer treatment, and for the short-term needs of future families of children receiving cancer treatment during times of increased health concerns (e.g., flu or RSV seasons). Families of children receiving treatment during the first year of the COVID-19 pandemic may require ongoing screening and mental health treatment given the multiple stressors they endured during this time. As they transition out of treatment and into survivorship care, it is crucial that they receive adequate support. Future families of pediatric cancer patients during times of public health crises may benefit from accessible, virtual mental health services available on short notice, in addition to continued assessment of family needs during unpredictable times. 

## 5. Conclusions

The current qualitative study sought to evaluate risk and resilience factors of families of pediatric cancer patients throughout the COVID-19 pandemic. Guided by Prime and colleagues’ theoretical framework, our findings suggest that these families have some experiences that aligned with the universal impacts of COVID-19 on families, but also have unique experiences. These families experienced financial burden due to cancer-related costs, and COVID-19 lead to additional burden. COVID-19 may lead to disruption in rules and routines, and the up-keep of these expectations is crucial for child adjustment in these particular families. These families may have strength during this time, as they may have been uniquely prepared for COVID-19 mandates, including handwashing and social distancing. Clear communication is a universal resilience factor for families during the pandemic and during cancer treatment. Families of the present study emphasize the importance not only of communication between family members but with the medical team as well. Interventions aimed at promoting resilience of families of pediatric cancer patients during the pandemic should be sure to address the unique needs of these families at this time. 

## Figures and Tables

**Table 1 ijerph-20-05208-t001:** Demographic Characteristics of Participating Caregivers (N = 31) and their Children Receiving Cancer Treatment.

Demographics	M (SD)	Percentage (%)	N
Caregiver			
Female sex		78	24
Marital Status			
Single		13	4
Married/Partnered		78	24
No response		9	7
Education Level (%)			
Finished high school or GED		3	1
Started college or trade school		22	7
Finished college or trade school		41	13
Finished masters or doctorate		16	5
No response		9	3
Child			
Age (years)	6.61 (4.21)		
Female sex		50	16
Race			
White/Caucasian		78	24
Black/African American		6	2
Hawaiian/Other Pacific Islander		6	2
Multi-Racial		3	1
No response		7	2
Non-Hispanic Ethnicity		91	28
Insurance Type			
Private		71	22
Medicaid		19	6
Government, COBRA, or Self-pay		10	3
Diagnosis			
Leukemia		56	17
Solid Tumor		16	5
Lymphoma		16	5
Brain/CNS Tumor		6	2
Other		6	2
Time since Diagnosis (months)	9.18 (6.48)		
Number of people living in home	4.90 (1.46)		

## Data Availability

The data that support the findings of this study are available on request from the corresponding author. The data are not publicly available due to privacy or ethical restrictions.

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
