# Peer review of "Risk and Resilient Functioning of Families of Children with Cancer during the COVID-19 Pandemic"

_ijerph, 2023, doi:10.3390/ijerph20065208_

Round 1
Reviewer 1 Report
Thank you for the opportunity to review this very interesting article. I believe that the adaptation of families of pediatric cancer patients during coronavirus restrictions is a highly relevant topic and I appreciate the effort with which the authors have addressed this topic. I have only a few minor comments on the paper.
Materials and methods
- Please specify the numbers of respondents participating in the study personally during clinic appointments and remotely via phone: given the Covid restrictions related to the access to health care, are there any differences between those who attended clinic appointments and those who participated remotely?
- Table 1: please provide numbers in addition to percentages. Please check/correct the column header name (“Percentile”).
- Given the predominance of women in the composition of your cohort (78% female caregivers), could you please consider the impact of this factor on the results in discussion/limitations?
Analytic plan
- I am not an expert in qualitative methodology but I would appreciate more details on creating categories described in results.
Results and Discussion
- At line 235, you mentioned that some families describe also positive impact of social disruption related to Covid. Could you please address these positive consequences (along with the negative ones) in more detail in Discussion to provide the reader with more complex view of patient experiences?
Reviewer 2 Report
The authors aimed to investigate the impact of the COVID-19 pandemic on families of pediatric cancer patients using qualitative methods. I think the article is generally well-written and the topic is significant to readers of IJERPH. I hope the authors find my feedback beneficial, organized by section below.
INTRODUCTION:
-The authors did a nice job of summarizing extant research clearly. They provided a nice justification for their study in this section.
-The sentence on pg 1, line 38-39 is awkward and would benefit from revision for succinctness or clarity
-The authors introduced Prime and colleagues conceptual framework beginning on line 73. Because this framework is so central to their purpose, I would recommend including a figure of the model.
-Related, because this framework has guided their study, I would recommend further developing their description of the framework. For example, the authors state "Prime and colleagues have identified familial processes and individual factors that may increase risk or promote resilience during the COVID-19 pandemic..." (lines85-87), but they did not discuss what those factors are. This would really help the reader better understand the aims of the study and provide more structure when discussing results
MATERIALS AND METHODS:
-The authors wrote that "The data presented in the current analysis were collected after the onset of the COVID-19 pandemic between..." (lines 126-127). Please clarify what data is being referred to here: were families interviewed during this time? Were they interviewed before the pandemic and then reapproached for COVID-specific questions?
-Please clarify what is meant by "The current sample is a subset of participating families who completed COVID-19 measures during the first year of the pandemic." What is meant by "COVID-19 measures"? (just the interview, or were other measures also given?) What is the subset based on? Participation?
-For Table 1, it would be helpful to also add the ns to percentiles
-It would be helpful to include the questions used in the semi-structured interviews (within the text or as a supplemental item)
-It would be helpful to clarify the analytic plan. For example, please directly state the pre-determined themes you used that were derived from Prime et al.
RESULTS:
-The clarity of the results section would be improved by categorizing themes by risk and resilience factors. The Prime et al. model used discussed a "mutually reinforcing system", but the connections between themes and family relationships is not apparent in these results.
-The clarity of the results section would also be improved by streamlining theme categories. For example, leniency with rules appeared under several theme designations (belief systems, organization, psychological/physical well being, etc.).
-Similarly, the themes and family responses vascilated between the specific impact of COVID-19 and to be diagnosed with cancer in general. Because we have a lot of research on the impact of a pediatric cancer diagnosis in families, this paper would be really strengthened by focusing on the specific impact of COVID-19. That would also bring a lot of clarity to the results.
-It would be helpful to provide more of a context to the family quotes used, either by providing more context in brackets within quotes or by providing more information when introducing quotes
-On lines 167-168, the authors reported that themes emerged. This is a bit confusing--are these "emerged" themes in addition to the pre-determined themes from Prime et al. that were previously mentioned? Please clarify.
-When reporting qualitative responses, sometimes it is still helpful to know what percentage of participants responded in this way (for example, X percent of X number of responses or X percent of families reported that organizational processes were disrupted). This helps give the reader a sense of how much "weight" this result carries--was it a theme from most of your participants or only a subset?
DISCUSSION:
-Similar to my recommendation above, I think this section would be significantly strengthened by sticking to results about the specific impact of COVID-19 rather than both the pandemic and general reactions to a cancer diagnosis. It is hard to really glean what impact the pandemic had because it is too interspersed with general difficulties stemming from a cancer diagnosis.
-In general, the unique contribution of this paper would be enhanced by considering what their families responses mean about the impact of the pandemic. For example, beginning on line 158 the authors stated that job loss and financial strain due to COVID-19 may increase financial difficulties for these families, but it is not clear what results they are basing this conclusion on. Further, beginning on line 492, the authors report that clear concrete information from their providers was helpful, but did not explain how COVID-19 impacted this.
-The authors stated that their study has implications for unique long term and short term needs for families during the pandemic (beginning line 519); however, their information is only from one time point during the pandemic. This statement should be revised.
-It is difficult to determine if the authors conclusions are supported by their data because of discussing both general and covid-19 specific implications. I would suggest the authors use their statement beginning on line 534 ("these families have some experiences that aligned with the universal impacts of COVID-19 on families, but also have unique experiences") and use that to structure their results and discussion section. In its current form, the authors did not adequately communicate the unique experiences of these families because of COVID.
-The authors may want to remove the section on Patents, as it seems this was included in error.
Round 2
Reviewer 2 Report
I appreciate the authors' attention to my comments. I think they have made important clarifications to the methods and results sections. A few additional considerations are recommended to further strengthen the paper:
-You really streamlined the results section by denoting risk and resiliency factors, as well as theme categories from Prime et al. However, your purpose statement is not consistent throughout (e.g., pg 3 lines 94 - 97 vs pg 15 lines 556 - 557). In the former, the purpose states "we aim to understand experiences of families that are consistent with the general impact of the pandemic on families across the US (regardless of child cancer diagnosis), and highlight unique impacts of the pandemic on families of children with cancer" while the latter states "The current qualitative study sought to evaluate risk and resilience factors of families of pediatric cancer patients throughout the COVID-19 pandemic." It seems the latter is more consistent with the authors' actual intent of the paper.
-I understand that including percentages of participant responses within each theme may be a matter of preference. The authors cited Miller and West's editorial (2014), but their recommendations are specific to the journal Addiction. I would encourage the authors to discuss preferences of IJERPH to determine whether or not to include percentages. As a reader, it is helpful for me to have this information to clarify patterns of data in themes and better understand key findings in themes. However, I defer to the editors.
